# Valve Endothelial Cell Exposure to High Levels of Flow Oscillations Exacerbates Valve Interstitial Cell Calcification

**DOI:** 10.3390/bioengineering9080393

**Published:** 2022-08-16

**Authors:** Chia-Pei Denise Hsu, Alexandra Tchir, Asad Mirza, Daniel Chaparro, Raul E. Herrera, Joshua D. Hutcheson, Sharan Ramaswamy

**Affiliations:** 1Department of Biomedical Engineering, Florida International University, Miami, FL 33199, USA; 2Miami Cardiac & Vascular Institute, Baptist Health South Florida, Miami, FL 33199, USA

**Keywords:** oscillatory flow, shear stress, valve calcification, paracrine signaling

## Abstract

The aortic valve facilitates unidirectional blood flow to the systemic circulation between the left cardiac ventricle and the aorta. The valve’s biomechanical function relies on thin leaflets to adequately open and close over the cardiac cycle. A monolayer of valve endothelial cells (VECs) resides on the outer surface of the aortic valve leaflet. Deeper within the leaflet are sublayers of valve interstitial cells (VICs). Valve tissue remodeling involves paracrine signaling between VECs and VICs. Aortic valve calcification can result from abnormal paracrine communication between these two cell types. VECs are known to respond to hemodynamic stimuli, and, specifically, flow abnormalities can induce VEC dysfunction. This dysfunction can subsequently change the phenotype of VICs, leading to aortic valve calcification. However, the relation between VEC-exposed flow oscillations under pulsatile flow to the progression of aortic valve calcification by VICs remains unknown. In this study, we quantified the level of flow oscillations that VECs were exposed to under dynamic culture and then immersed VICs in VEC-conditioned media. We found that VIC-induced calcification was augmented under maximum flow oscillations, wherein the flow was fully forward for half the cardiac cycle period and fully reversed for the other half. We were able to computationally correlate this finding to specific regions of the aortic valve that experience relatively high flow oscillations and that have been shown to be associated with severe calcified deposits. These findings establish a basis for future investigations on engineering calcified human valve tissues and its potential for therapeutic discovery of aortic valve calcification.

## 1. Introduction

Calcific aortic valve disease (CAVD), one of the most prevalent chronic heart problems, involves hardening of aortic valve leaflets due to calcium phosphate deposition, resulting in stenosis, regurgitation, and reduced cardiac output. Clinical data have shown a global increase in over 100% of CAVD cases in the past 30 years [1]. Current treatment options for early CAVD intervention are not available, and the main factors associated with development of CAVD remain unclear. Heart valves are subject to various mechanical stresses from hemodynamic flow environments, and heart valve remodeling occurs in response to cyclic mechanical loading [2]. Healthy cardiac function requires the aortic heart valve to facilitate unidirectional flow from the left ventricle into the aorta to the systemic circulation during systole, with minimal blood flow resistance.

Most aortic valves consist of three equal-sized leaflets forming three lines of coaptation. Bicuspid aortic valve malformations are some of the most common congenital heart conditions that affect approximately 1–2% of the population [3]. Each valve leaflet contains a ventricularis layer, a spongiosa layer, and a fibrosa layer (Figure 1). The ventricularis consists of mainly elastin fibers and faces the left ventricle. The fibrosa layer is mainly composed of collagen and faces the aortic side of the heart. The spongiosa layer is sandwiched between the ventricularis and fibrosa layers and is mainly composed of glycosaminoglycans [4]. Due to the valve structure and direction of blood flow, laminar flow with high shear stress is mainly observed on the ventricularis side, while the fibrosa layers are mainly dominated by low shear stress and oscillatory flow [5]. CAVD mineral deposition is most often clinically observed on the fibrosa layer.

Studies have shown that low shear stresses are commonly associated with vascular lesions and calcifications [6,7]. Specifically, regions of valve tissues that experience low wall shear stress, coupled with blood flow oscillations, triggers inflammation [8]. This pathological mechanical environment is present on the heart valve’s fibrosa layer [9]. To further specify and quantify flow disturbances to valve calcification, we utilized the oscillatory shear index (OSI) as a parameter to correlate precise flow oscillation magnitudes with the development of CAVD. OSI is a measurement of flow disturbances that quantifies the ratio between the forward flow net temporal shear stress to the total temporal shear stress magnitude that is assumed to be always positive (Equation (1)), and the OSI value ranges between zero (no oscillation, or steady flow) to 0.50 (full oscillation, or forward flow in half the temporal cycle and reversed flow in the other half) [10]. Using OSI as a quantitative description of flow oscillations to connect to valve calcification has not been previously investigated. If there is an association between OSI and heart valve calcification, this can subsequently be used to create a human calcific valve engineered tissue model system to assess emerging therapeutics to treat calcific valve disease. As a first step, we therefore examined VIC responses to the paracrine signaling of biochemical end-products from VECs that were exposed to varying OSI values.
(1)$$\mathrm{OSI}=\frac{1}{2}\left(1-\frac{\left|\;\int_{0}^{T}\tau_{w\;dt}\right|}{\int_{0}^{T}\left|\tau_{w\;dt}\right|}\;\right)$$**Equation (1).** Oscillatory shear index equation, where *T* = duration of cycle, *t* = time, and *τ_w_* = wall shear stress [10].

## 2. Materials and Methods

### 2.1. Computational Fluid Dynamics (CFD)

A CFD simulation of a patient with CAVD was conducted to investigate the role of fluid oscillations on the diseased state. A digital model of a human heart valve in the early diastolic phase of an 82-year-old female patient with CAVD (Figure 2A) was commercially obtained (Valve-012-Heart Print catalog, Materialise, Plymouth, MI, USA). The calcification regions were removed from the surrounding tissues using ANSYS Spaceclaim (Ansys Inc., Canonsburg, PA, USA) to approximate the original healthy valve geometry. A finite element analysis (FEA) was performed in ANSYS Mechanical to convert the closed healthy geometry to its systolic equivalent with a ventricular pressure of 120 mmHg (Figure 2B). The valve leaflets were modeled as three-parameter incompressible Mooney–Rivlin material, the surrounding sinuses were treated as isotropic linear elastic, and the calcification regions were treated as a first order Ogden model [11]. The healthy geometry was then meshed and used for a pulsatile flow-based CFD simulation. This CFD simulation assumed a constant outlet aortic pressure of 100 mmHg and used a physiologically relevant inlet blood velocity waveform, which we have previously reported on [12]. Blood was modeled as a non-Newtonian fluid using the Carreau model [12,13]. Flow oscillations were then quantified using the OSI parameter.

### 2.2. Analyses of Excised Aortic Valve Leaflets

De-identified, excised calcified human aortic heart valves were obtained from the Miami Cardiac and Vascular Institute (MCVI). The study was approved by the Baptist Health South Florida Institutional Review Board (IRB) under study number 1189342, and all procedures were conducted according to their guidelines. Consent was obtained from each patient prior to their enrollment into the study. Each valve was transported as a de-identified sample and in a saline medium, from the hospital operating room to the lab, immediately after aortic valve replacement surgery. The valves were photographed on both the fibrosa and ventricularis layers using a digital camera.

### 2.3. In Vitro Cell Culture Experiments and Subsequent Assessments

Rat (species: *Rattus norvegicus*) VECs and VICs were purchased from Celprogen, Inc. (Torrance, CA, USA) and Innoprot (Bizkaia, Spain), respectively. The VECs were expanded in extracellular matrix-coated T75 culture flasks (Celprogen, Inc.) with rat valvular endothelial primary cell culture complete growth media with serum and antibiotics (Celprogen, Inc.), and the VICs were expanded with Dulbecco’s modified Eagle medium (DMEM) containing 10% FBS and 1% P/S in non-coated T75 culture flasks. VECs were then seeded with gelatin at 2.0 × 10^5^ cells per channel in a 24-well Bioflux plate consisting of 8 microfluidic channels per plate (Fluxion Biosciences, Inc., Alameda, CA, USA) using DMEM containing 10% FBS and 1% P/S. Upon 24 h after seeding and confirmation of VEC attachment, the VECs were then conditioned for 48 h in a shear stress cell assay system (Bioflux, Fluxion Biosciences, Inc., Alameda, CA, USA) at an average shear stress magnitude of 1 dyne/cm^2^ to promote an atherogenic environment [14]. Each Bioflux well plate with the cells was conditioned under an OSI flow group for 48 h, and a total of four flow groups was investigated: static (0 OSI/no flow), steady flow (0 OSI/steady flow), 0.25 OSI (moderate oscillation), and 0.50 OSI (full oscillation). Conditioned media from all four VEC flow groups were collected separately from each Bioflux plate, and an equal volume of pro-calcifying (PC) media was added to each of the collected VEC-conditioned media groups. The final VEC-conditioned PC media mixture consisted of 1.8 mM CaCl_2_ (Sigma-Aldrich, St. Louis, MO, USA), 3.8 mM NaH_2_PO_4_ (Sigma-Aldrich), 0.4 units/mL of inorganic pyrophosphate (Sigma-Aldrich) [15], and 5% FBS with 1% P/S [16]. The VEC-conditioned PC media was then subsequently used to statically culture VICs in 12-well plates for 7 days, with one media change that was performed on day 4 for the respective VEC-conditioned flow groups. Upon termination of VIC exposure to media from various VEC-conditioned flow groups, VIC calcification was measured using alizarin red staining (ARS). The alizarin red dye was then extracted and quantified with a microplate reader at 405 nm (Synergy HTX Multimode Reader, Biotek Agilent, Santa Clara, CA, USA). Three replicates were conducted for each conditioning group, and data were evaluated using a one-way ANOVA in conjunction with Tukey’s post hoc analysis in SPSS (IBM, Armonk, NY, USA) with statistical significance identified when *p* < 0.05. Key phenotypic markers expressed by VICs conditioned in various VEC-paracrine communicated flow groups were also assessed using real time quantitative polymerase chain reaction (RT-qPCR) at three replicates per target gene per flow group. Data from RT-qPCR consisted of cycle threshold, or CT values, which were analyzed using the Livak method ΔΔCT [17] to compute fold change with Fresh PC as the control group and Actb as the housekeeping gene [18]. The Fresh PC media consisted of only pro-calcifying ingredients with no paracrine signaling from VECs.

## 3. Results

### 3.1. CFD

CFD results from the CAVD affiliated patient showed various OSI values in the original healthy aortic configuration (Figure 2C) at peak systole. Largest oscillations of 0.50 OSI were found on the non-coronary cusp, which correlated with the original sites of calcification from the parent geometry before computational removal of calcific regions.

### 3.2. Analyses of Excised Aortic Valve Leaflets

Images from clinical observations showed severe mineral deposition on the fibrosa surfaces compared to their respective ventricularis surfaces (Figure 3). Regions of brownish and reddish colors indicated valve mineralization, which were mainly found around the annuli or the cusps of the valves on the fibrosa surfaces. However, the respective ventricularis surfaces maintained an overall healthier tissue layer with lighter color. 

### 3.3. In Vitro Cell Culture Experiments and Subsequent Assessments

The ARS results revealed the highest VIC calcification in the 0.50 OSI group (Figure 4). Specifically, statistical assessments showed significantly increased calcification in the 0.50 OSI group compared to the 0 OSI Static (*p* < 0.05), 0 OSI Steady (*p* < 0.05), and 0.25 OSI (*p* < 0.05) groups. Comparisons of VIC calcification between Fresh PC vs. 0 OSI Static and 0 OSI Steady vs. 0 OSI Static were also significantly different (*p* < 0.05), while VIC calcification between 0.25 OSI vs. 0 OSI Static and 0.25 OSI vs. 0 OSI Steady groups were not significantly different (*p* > 0.05). VIC calcification between Fresh PC and 0.50 OSI groups were also not statistically significant (*p* > 0.05). 

Gene expression results (Figure 5) indicated upregulation in calcific genes in the Fresh PC and 0.50 OSI groups, specifically Runx2, Mmp2, Tnap, and Bmp2. The highest alpha-SMA expression was also observed in the Fresh PC group, and the highest expression of type I collagen was observed in the 0.25 OSI group. However, the fold changes were not statistically significant (*p* > 0.05) across the groups.

## 4. Discussion

CAVD is an ongoing chronic condition that affects over 100,000 people worldwide annually, with an observed incidence rate increase from 3.25/100,000 persons to 7.13/100,000 persons between 1990 and 2019 [19]. As early treatment is currently unavailable mainly due to the lack of understanding of CAVD progression, the rising global incidence calls for finding possible alternatives for early CAVD interventions in the healthcare industry. We know that clinical observations have shown valve mineralization specifically on the fibrosa layer of the valve [20], and that the endothelium on the fibrosa side is subject to oscillatory flow [21]. This suggests a possible linkage between flow oscillations and valve calcification via changes in fibrosa phenotypes caused by hemodynamic factors. Understanding this mechanobiology may enable further developments in early drug intervention.

Our findings through CFD analysis (Figure 2) suggests that in the original healthy valve configuration, high OSI is mainly observed in the fibrosa regions where calcific plaques were formed. In the CFD analysis, the NCC cusp was the most diseased compared to LCC and RCC. This correlates to a lower shear stress and high flow oscillation on NCC compared to other cusps, resulting in calcific lesions [22,23]. The images of the explanted severely calcified valves (Figure 3) on the fibrosa layer concur with previous studies that show the valve fibrosa layer being most critically involved in aortic valve calcification, specifically near the annulus, or base regions [8,24]. As the hemodynamic environments between the ventricularis and fibrosa sides differ in both flow oscillations and shear stress magnitudes, the difference in VEC linings on ventricularis and fibrosa surfaces due to mechanical environments may be primarily responsible for side-specific valve diseases such as CAVD [25]. It is known that immune cells may enter the valve in response to valve endothelium injury, followed by proliferation of myofibroblast-like cells on the fibrosa layer [26]. These myofibroblasts eventually differentiate into osteoblast-like cells, which results in valve calcification [27]. The base regions of valve leaflets are also subject to the lowest shear stress and highest OSI compared to the belly or tip regions [28], which in turn make the base regions more vulnerable to lesions and plaque formation [14].

The in vitro findings in our present study show that a combination of pro-calcifying media with high OSI (OSI = 0.50) significantly increases (*p* < 0.05) VIC calcification via VEC paracrine signaling. This corroborates current theories of increased VIC calcification under disturbed flow [9], in addition to clinical observations of increased mineral deposition occurring on the fibrosa surfaces of valve leaflets, where regions of flow oscillations are mostly observed. These in vitro findings also further confirm our CFD results, in which regions with higher OSIs mapped in the peak systole configuration were more susceptible to mineralization. In the pro-calcifying VEC-conditioned media samples (Figure 5), the presence of paracrine regulation from VECs seems to generally lower gene expressions associated with calcification or the osteogenic phenotype in VICs; specifically, Tnap, Bmp2, Runx2. Runx2, and Bmp2 are commonly associated with osteogenic differentiation of VICs [29], whereas αSMA may be associated with increased cell contractile activity and wound healing [30,31]. The 0 OSI Static group also exhibited an increase in Bmp2 expression, and this may be due to extremely low, or zero shear stress that is known to be associated with calcification in vascular systems [14]. This observation is like previous studies of VECs attenuating aortic valve diseases, and that dysfunction of the valve endothelium can initiate VIC calcification [32]. In addition, the 0.25 OSI group exhibited a general downregulation of calcific genes such as Bmp2, Mmp2, and Tnap, as well as upregulation of Col1a1, which produces collagen fibers that are densely found in the fibrosa extracellular matrix. This concurs with previous studies in which physiologically relevant moderate flow oscillations could promote favorable phenotypic expressions in maintaining valve tissue integrity [33].

These findings suggest that VECs exposed to low-to-moderate levels of flow oscillations maintain a quiescent VIC phenotype via paracrine signaling. On the other hand, pro-calcific stimuli coupled with high oscillatory flow regions (OSI = 0.50) on VECs lead to substantial risk of increased VIC calcification. This is the major finding of our current study, which showed that the regions of high oscillations at an OSI = 0.50 directly associate with VEC–VIC paracrine communications under pro-calcific biochemical environments to induce aortic valve calcification (Figure 4). The increased calcification tendencies are also reflected in the relatively higher expression of calcific genes such as Bmp2 and Mmp2 in the 0.50 OSI group (Figure 5). Hence, in pro-calcific bio-chemical environments, high levels of flow oscillation are the specific flow disturbances that induce the build-up of calcified valve deposits. However, whether molecular cues in this VEC-to-VIC paracrine regulated pathway can be targeted to reduce valve calcification requires further investigation. Our current findings specifically were based on VIC culture in the VEC-conditioned media, which consisted of both VEC-released cytokines, non-exosomes, and exosome secretions. Some limitations include not using human cells for the study and not fully recapitulating the valve anatomy, as VECs and VICs co-exist in valve tissues in a 3-dimensional setting. Another limitation of our study is with regards to our CFD analysis. The valve was commercially acquired and was already provided in its calcified state. Therefore, we did not have access to an actual healthy valve geometry. Hence, the “healthy valve” simulation was done by artificially removing the calcified deposits computationally, to mimic what the healthy valve may have looked like prior to its calcification. However, we did attempt to substantiate this limitation by confirming that the hemodynamics of the healthy valve simulation, in terms of its pressure gradient (ΔP) and peak velocity, matched those reported clinically for a healthy valve [34]. The study also does not assume or assess any co-existing chronic health conditions, such as diabetes, that can also influence the level of valve calcification. In addition, the calcific gene expressions did not show statistically significant upregulation in the 0.50 OSI group. Future studies will involve uncovering possible pathways that are primarily responsible for inducing calcification by the VICs at the tissue level. These future studies will emphasize the development of a 3-dimensional valve calcification engineered tissue model system with co-cultures of VECs and VICs seeded on scaffolds.

## Figures and Tables

**Figure 1 bioengineering-09-00393-f001:**
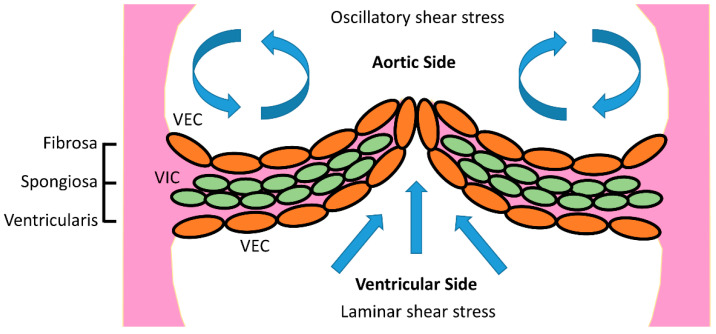
Anatomy of an aortic heart valve. Flow oscillations are commonly found in the fibrosa layer, while the ventricularis layer is mainly subject to laminar flow.

**Figure 2 bioengineering-09-00393-f002:**
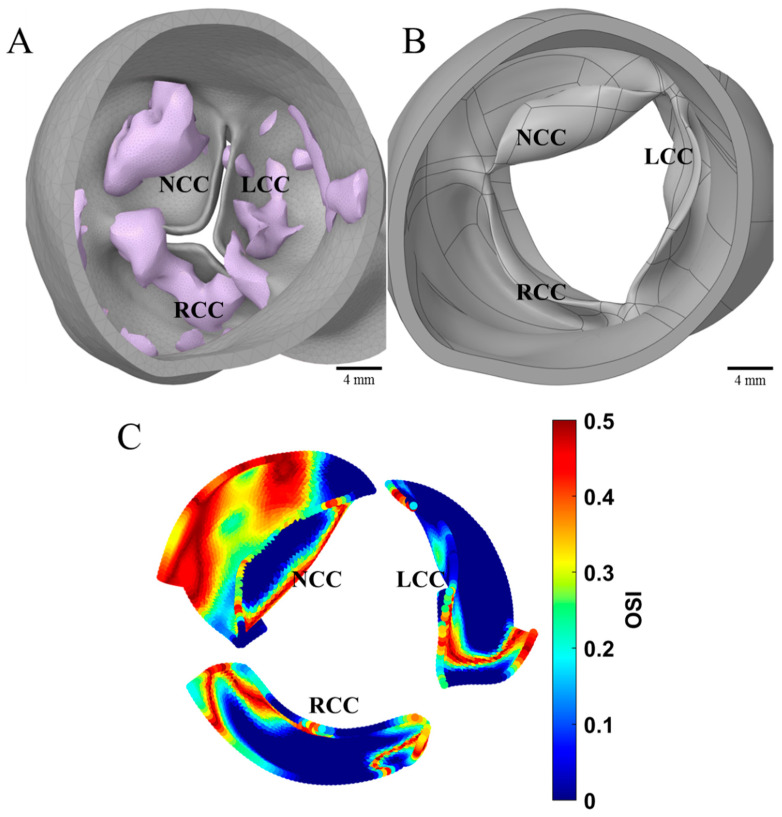
(**A**) Original diastolic configuration of diseased valve with tissue and calcification (Valve-012-Heart Print catalog, Materialise, Plymouth, MI, USA). (**B**) Healthy valve at peak systole after calcification removal and FEA simulation. (**C**) CFD simulation of OSI contours on healthy valve at peak systole. LCC, NCC, RCC: Left, non, and right coronary artery cusps, respectively.

**Figure 3 bioengineering-09-00393-f003:**
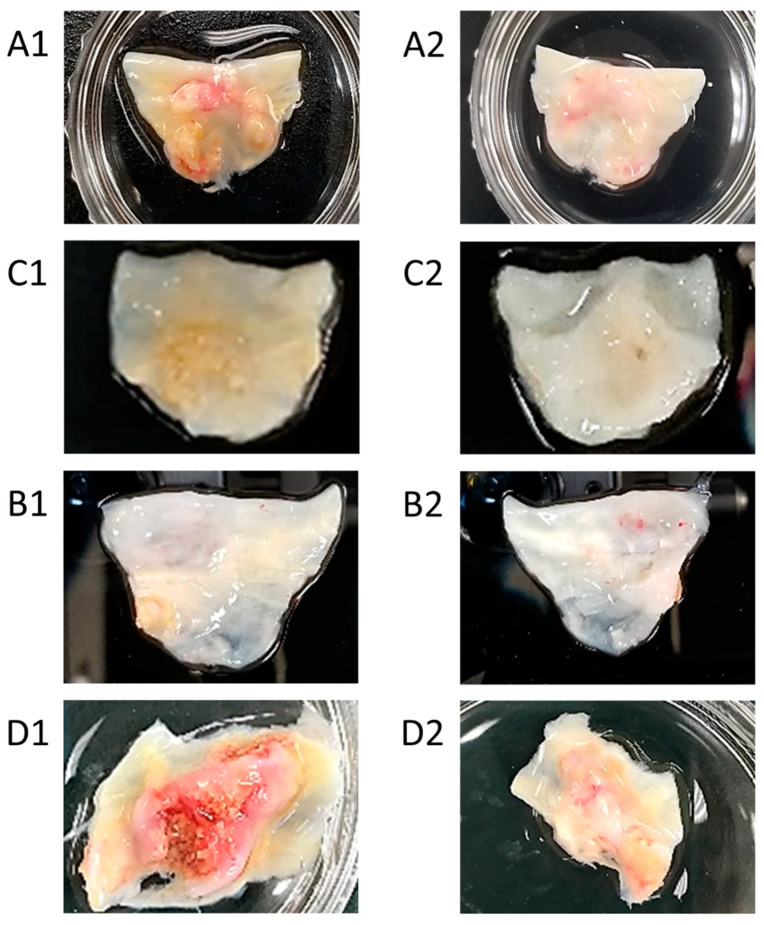
(**A1**) Fibrosa and (**A2**) ventricularis layers of explanted calcified aortic valves from patient 1. (**B1**) Fibrosa and (**B2**) ventricularis layers of explanted calcified aortic valves from patient 2. (**C1**) Fibrosa and (**C2**) ventricularis layers of explanted calcified aortic valves from patient 3. (**D1**) Fibrosa and (**D2**) ventricularis layers of explanted calcified aortic valves from patient 4. Clear evidence of substantially more calcification was present on the fibrosa side of the aortic valve.

**Figure 4 bioengineering-09-00393-f004:**
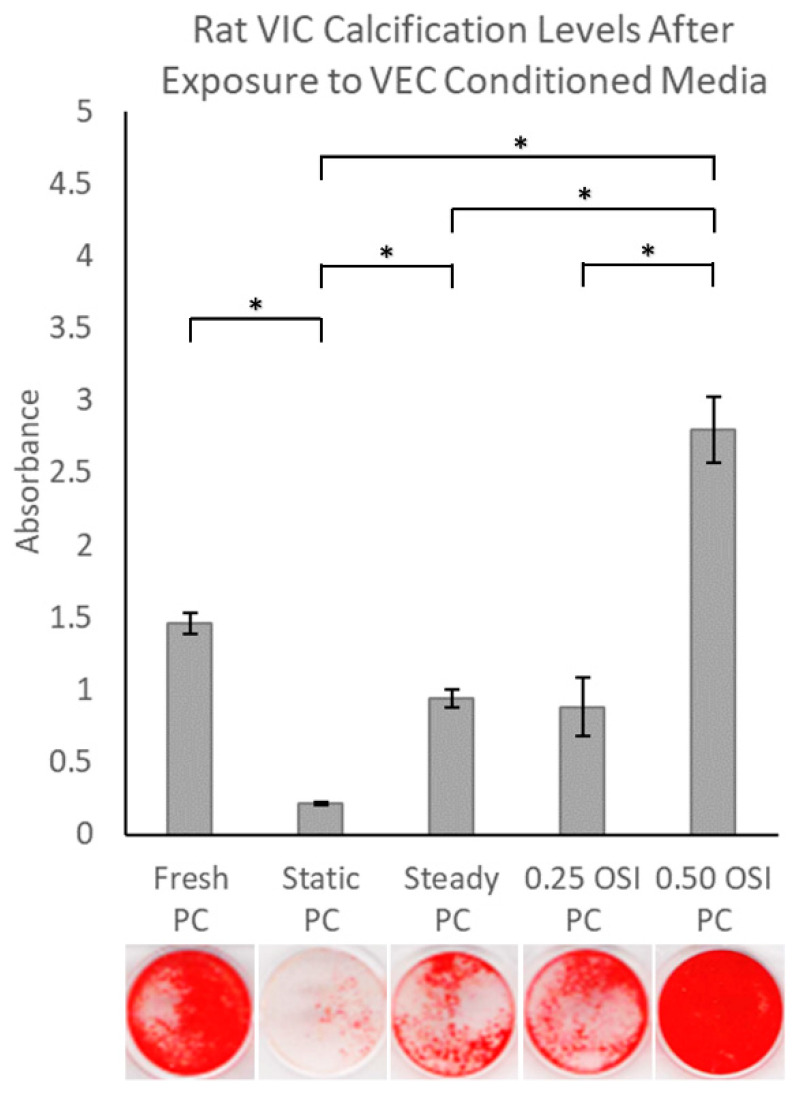
Quantified ARS of VIC calcification in oscillatory flow-conditioned VEC media with pro-calcifying components. * Statistical significance, *p* < 0.05.

**Figure 5 bioengineering-09-00393-f005:**
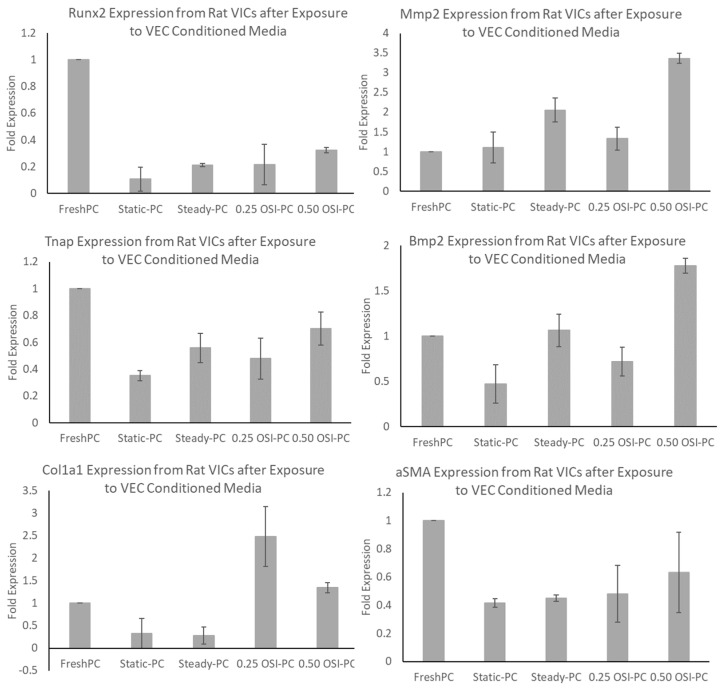
Rat VIC gene expression after 7-day exposure to pro-calcifying VEC-conditioned media.

## Data Availability

Access to data reported in this manuscript can be requested by e-mailing the corresponding author Sharan Ramaswamy at: sramaswa@fiu.edu.

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
