# Peer review of "Valve Endothelial Cell Exposure to High Levels of Flow Oscillations Exacerbates Valve Interstitial Cell Calcification"

_bioengineering, 2022, doi:10.3390/bioengineering9080393_

Round 1

Reviewer 1 Report

The article entitled, “Valve Endothelial Cell Exposure to High-Levels of Flow Oscillations Exacerbates Valve Interstitial Cell Calcification,” by Hsu et. al. addresses an important aspect of mechanobiology of heart valves that has not previously been explored: oscillatory shear index (OSI). As the authors clearly identified in the first several figures, calcific aortic valve disease is highly correlated with OSI on differing sides of the valve and understanding this correlation better will allow for a maturation of a treatment development-cantered model. The experimental evidence collected attempts to determine how shear flow over VECs leads to the development of a calcific and/or pro-fibrotic phenotype in VICs through paracrine signalling.

The use of conditioned media is a good approach for analysing this question. However, this paper does need some major edits which all centre around the use of media types that do not protect cells from developing a diabetic phenotype. This is because VECs are notorious for becoming diabetic under high glucose or high insulin levels. In particular, please address these concerns:

1.     At a minimum, please run control groups of the static PC and fresh PC using a low-glucose media such as low-glucose DMEM or Rat Valvular Endothelial Primary Cell Culture Complete Growth Media to determine if the high glucose content may be affecting the diabetic profile of the cells. If differences are found in these new control groups and the groups you have already done, then the text will need to be re-written with diabetes in mind. Depending on the results from these new runs, the other conditions may need to be re-done using the low-glucose media as well.

2.     10% FBS was used throughout the experiment. Insulin in FBS is well known to activate transcription of Runx2, and MMPs and BMPs; especially in a diabetic setting. Please repeat the experiment using 0% (or 1% if you have difficulty with the cell viability) FBS serum starvation to limit exogenous insulin-induced activation of the transcripts you are measuring. This way you will know that the exogenous signalling is coming from VEC paracrine factors and not the FBS. Keep with the low-glucose media during these experiments, too.

In addition to the above major concerns, there are a few minor concerns as well.

1.     1 dyne/cm2 is quite low and may not be physiologically relevant, but this may be a limitation of the instrumentation. If a more physiological value of shear can be used in the device, please consider using it.

2.     Please indicate what ECM molecule(s) were used for coating, as this affects the expression level of alpha-SMA in VICs considerably.

3.     Please indicate what post-test was used in conjunction with the one-way ANOVAs.

Author Response

REVIEWER 1:

Comments and Suggestions for Authors:

The article entitled, “Valve Endothelial Cell Exposure to High-Levels of Flow Oscillations Exacerbates Valve Interstitial Cell Calcification,” by Hsu et. al. addresses an important aspect of mechanobiology of heart valves that has not previously been explored: oscillatory shear index (OSI). As the authors clearly identified in the first several figures, calcific aortic valve disease is highly correlated with OSI on differing sides of the valve and understanding this correlation better will allow for a maturation of a treatment development-cantered model. The experimental evidence collected attempts to determine how shear flow over VECs leads to the development of a calcific and/or pro-fibrotic phenotype in VICs through paracrine signalling.

The use of conditioned media is a good approach for analysing this question. However, this paper does need some major edits which all centre around the use of media types that do not protect cells from developing a diabetic phenotype. This is because VECs are notorious for becoming diabetic under high glucose or high insulin levels. In particular, please address these concerns:

  1. At a minimum, please run control groups of the static PC and fresh PC using a low-glucose media such as low-glucose DMEM or Rat Valvular Endothelial Primary Cell Culture Complete Growth Media to determine if the high glucose content may be affecting the diabetic profile of the cells. If differences are found in these new control groups and the groups you have already done, then the text will need to be re-written with diabetes in mind. Depending on the results from these new runs, the other conditions may need to be re-done using the low-glucose media as well.

RESPONSE: Thank you for the suggestion. While a diabetic VIC phenotype will increase the probability of VIC calcification, all VICs in our study were exposed to the same level of glucose concentration for all groups, i.e., Static PC, Fresh PC, Steady PC, 0.25 OSI PC, and 0.50 OSI PC. Adding a different control group with low glucose to the Static PC and Fresh PC would introduce another variable on top of our OSI variable. Also, we are unable to track the diabetic conditions of the de-identified patients’ calcific valves in the manuscript to rule out or confirm diabetes as a factor for their VIC calcification. Therefore, in order to objective with our conclusions, our focus has to remain on having the OSI as the only variable factor between the groups, specifically that assesses the level of VIC-induced calcification under the different OSI flow profiles, with everything else being the same between the groups, including the glucose concentration. We do agree with you completely a thank you again for reminding us that concomitant pathologies such as diabetes can lead to different outcomes.  We have therefore added a limitation stating that the study conducted does not assume any chronic conditions that may influence valve calcification, in the DISCUSSION section of the revised manuscript as follows:

“Some limitations include not using human cells for the study and not fully recapitulating the valve anatomy, as VECs and VICs co-exist in valve tissues in a 3-dimensional setting. The study also does not assume or assess any co-existing chronic health conditions, such as diabetes, that can also influence the level of valve calcification.”

  1. 10% FBS was used throughout the experiment. Insulin in FBS is well known to activate transcription of Runx2, and MMPs and BMPs; especially in a diabetic setting. Please repeat the experiment using 0% (or 1% if you have difficulty with the cell viability) FBS serum starvation to limit exogenous insulin-induced activation of the transcripts you are measuring. This way you will know that the exogenous signalling is coming from VEC paracrine factors and not the FBS. Keep with the low-glucose media during these experiments, too.

RESPONSE: Thank you for the comment. The gene expressions (Runx2, MMP, and BMP) were only conducted on human VICs. When conditioning human VICs using media collected from the Bioflux shear assay system, the FBS concentration was 5%, not 10% (section 2.3, line 126 in manuscript). The 5% FBS concentration is relatively low and was based on citation [16] S. Goto et al on FBS concentration to utilize for osteogenic media and pro-calcifying media.

In addition to the above major concerns, there are a few minor concerns as well.

  1. 1 dyne/cm2 is quite low and may not be physiologically relevant, but this may be a limitation of the instrumentation. If a more physiological value of shear can be used in the device, please consider using it.

RESPONSE: Thank you for the comment. You are correct that the average shear stress on the aortic heart valve is closer to 4 dynes/cm2.  However, regions on the fibrosa side of the valve are subjected to much lower shear stress, such as 1 dyne/cm2.  As these regions are also prone to calcification, our application of 1 dynes/cm2 is physiologically relevant in this regard.  Moreover, the 1 dyne/cm2 was also based on the citation [14] Malek et al, which states that atherogenic setting is < 4 dyne/cm^2. Since we are inducing VIC calcification, we would need a setting that is prone to disease phenotype. In order to clarify our use of a wall shear stress magnitude of 1 dynes/cm2, the following changes have been made to the MATERIALS AND METHODS section 2.3 in the revised manuscript:

“Upon 24 hours after seeding and confirmation of VEC attachment, the VECs were then conditioned for 48 hours in a shear stress cell assay system (Bioflux, Fluxion Biosciences, Inc. Alameda, CA) at an average shear stress magnitude of 1 dyne/cm2 to promote an atherogenic environment [14].”

  1. Please indicate what ECM molecule(s) were used for coating, as this affects the expression level of alpha-SMA in VICs considerably.

RESPONSE: Thank you for this important reminder. Gelatin was used in VEC during seeding in the Bioflux shear assay system. The VICs were not coated. The following changes have been made to the MATERIALS AND METHODS section 2.3, in the revised manuscript to indicate the use of non-coated flasks for VICs:

“The VECs were expanded in extracellular matrix-coated T75 culture flasks (Celprogen, Inc.) with Rat Valvular Endothelial Primary Cell Culture Complete Growth Media with Serum and Antibiotics (Celprogen, Inc.), and the VICs were expanded with Dulbecco’s Modified Eagle Medium (DMEM) containing 10% FBS and 1% P/S in non-coated T75 culture flasks.”

“VECs were then seeded with gelatin at 2.0 x 105 cells per channel in a 24-well Bioflux plate consisting of 8 microfluidic channels per plate (Fluxion Biosciences, Inc. Alameda, CA) using DMEM containing 10% FBS and 1% P/S.”

  1. Please indicate what post-test was used in conjunction with the one-way ANOVAs.

RESPONSE: Thank you for the reminder. We used the Tukey post-hoc test and have now added this to the revised manuscript, under the MATERIALS AND METHODS section 2.3 as follows:

“Three replicates were conducted for each conditioning group, and data was evaluated using a one-way ANOVA in conjunction with the Tukey’s post-hoc analysis in SPSS (IBM, Armonk, NY) with statistical significance identified when p<0.05.”

Reviewer 2 Report

This is a nicely written paper that attempts to relate flow oscillatory shear to  modulation of interstitial cell calcification pathobiology in the development of calcific aortic valve disease (CAVD).  Overall this is a novel approach to suggest cause-and-effect in an important pathological process for which has been difficult to understand mechanisms.

There is one key issue that begs explanation by the authors.  CAVD causes underlying structural damage to the valve.  The authors assume that computationally "decalcifying" the valve yields CFD simulation of a valve with "healthy geometry".  This is unlikely to be the case.  The authors should discuss their choice and explain why a normal valve was not used in the CFD simulation.  If this cannot be scientifically justified, then this assumption should be discussed as a potential limitation to the study results.

Author Response

REVIEWER 2:

Comments and Suggestions for Authors:

This is a nicely written paper that attempts to relate flow oscillatory shear to modulation of interstitial cell calcification pathobiology in the development of calcific aortic valve disease (CAVD).  Overall, this is a novel approach to suggest cause-and-effect in an important pathological process for which has been difficult to understand mechanisms.

There is one key issue that begs explanation by the authors.  CAVD causes underlying structural damage to the valve.  The authors assume that computationally "decalcifying" the valve yields CFD simulation of a valve with "healthy geometry".  This is unlikely to be the case.  The authors should discuss their choice and explain why a normal valve was not used in the CFD simulation.  If this cannot be scientifically justified, then this assumption should be discussed as a potential limitation to the study results.

RESPONSE:

We thank the Reviewer for this important suggestion. The following has now been added to the limitation portion of the DISCUSSION section in the revised manuscript:

“Another limitation of our study is with regards to our CFD analysis. The valve was commercially acquired and was already provided in its calcified state. Therefore, we did not have access to an actual healthy valve geometry. Hence, the “healthy valve” simulation was done by artificially removing the calcified deposits computationally, to mimic what the healthy valve may have looked like prior to its calcification. However, we did attempt to substantiate this limitation by confirming that the hemodynamics of the healthy valve simulation, in terms of its pressure gradient () and peak velocity, matched those reported clinically for a healthy valve [34].”

  1. Otto, C. M., and Prendergast, B., 2014, “Aortic-Valve Stenosis — From
    Patients at Risk to Severe Valve Obstruction,” New Engl. J. Med., 371(8), pp.
    744–756.
